# Advanced Fungal Biotechnologies in Accomplishing Sustainable Development Goals (SDGs): What Do We Know and What Comes Next?

**DOI:** 10.3390/jof10070506

**Published:** 2024-07-22

**Authors:** Pragya Tiwari, Kyeung-Il Park

**Affiliations:** Department of Horticulture & Life Science, Yeungnam University, Gyeongsan 38541, Republic of Korea; pki0217@yu.ac.kr

**Keywords:** bio-based economy, climate change, fungal biotechnologies, global food security, SDGs, ‘Wood Wide Web’

## Abstract

The present era has witnessed an unprecedented scenario with extreme climate changes, depleting natural resources and rising global food demands and its widespread societal impact. From providing bio-based resources to fulfilling socio-economic necessities, tackling environmental challenges, and ecosystem restoration, microbes exist as integral members of the ecosystem and influence human lives. Microbes demonstrate remarkable potential to adapt and thrive in climatic variations and extreme niches and promote environmental sustainability. It is important to mention that advances in fungal biotechnologies have opened new avenues and significantly contributed to improving human lives through addressing socio-economic challenges. Microbe-based sustainable innovations would likely contribute to the United Nations sustainable development goals (SDGs) by providing affordable energy (use of agro-industrial waste by microbial conversions), reducing economic burdens/affordable living conditions (new opportunities by the creation of bio-based industries for a sustainable living), tackling climatic changes (use of sustainable alternative fuels for reducing carbon footprints), conserving marine life (production of microbe-based bioplastics for safer marine life) and poverty reduction (microbial products), among other microbe-mediated approaches. The article highlights the emerging trends and future directions into how fungal biotechnologies can provide feasible and sustainable solutions to achieve SDGs and address global issues.

## 1. Introduction

The rising global population, climatic perturbations, and exhausting natural resources are key drivers of ecological imbalance and extinction of plants and animals. In the face of widespread damages and climatic uncertainties, existing biodiversity can support mankind and address the current challenges associated with providing bio-based resources and tackling environmental challenges, ecosystem restoration, and addressing global food demands [1,2,3]. Microorganisms exist as integral members of the ecosystem, demonstrating ubiquitous presence. Due to their significant association with and influence on human lives, microbes demonstrate remarkable potential to adapt and thrive in climatic variations and extreme niches and promote environmental sustainability [4]. Among other biological species, fungi comprise an integral component of our biodiversity and are estimated to include 2.2 and 3.8 million species [5]; however, the vast majority remain unexplored due to limited knowledge/insights about fungal biology and sophisticated technologies. The era of fungal biotechnology started with citric acid production (by controlled fermentation of *Aspergillus niger*) by Pfizer in 1919 and has expanded to commercial use in food additives and the chemical and pharmaceutical sectors [6,7]. Worldwide companies like Bayer, DuPont, Kerry Group, AB Enzymes, etc., are harnessing fungal resources for economic purposes. Several species of edible fungi are being extensively investigated as attractive resources of ‘high-value’ metabolites including antibiotics, food ingredients/additives, chemicals, industrial enzymes, pigments, etc. [8,9]. While filamentous fungi have been widely explored and harnessed, edible mushrooms (from *Ascomycota* and *Basidiomycota*), *Saccharomyces cerevisiae*, *Pichia pastoris*, and *Yarrowia lipolytica*, have been increasingly exploited for commercial use. The advances in fungal biotechnology have opened new avenues and significantly contributed to creating engineered strains with high product yields, bio-functionality, and value addition [10]. For the discovery of new/novel transformative medicines, an insightful discussion suggested that fungi have evolved to create genetically encoded small molecules (GEMs) that can be effective against human targets, and tend to have better pharmacokinetics– brain penetration, oral bioavailability, and less off-target effects, compared to synthetic agents [11] facilitated by advanced high-throughput technologies [12]. With considerable progress in omics biology and their integrated use, a vast repertoire of natural products has been identified and biologically evaluated, attributed to the recent insights on the biosynthetic pathways/mechanisms. Furthermore, optimized production of these compounds can be achieved in cultures via cultivation and metabolic methods including CRISPR-Cas9-mediated gene editing, metabolic engineering, and gene silencing [13]. The publicly available genome resources for fungal species *Trichoderma* spp., *Aspergillus* spp., *Ganoderma lucidum*, *Penicillium* spp., *Rhizopus* spp., and others [14,15] have opened new avenues in bridging knowledge gaps in fungal biology and biotechnologies. The advanced molecular predictions have considerably expanded the metabolic pool of fungal high-value metabolites and utilization for creating a bio-based economy and achieving SDGs. Through its policies and reforms, the United Nations SDGs aim to improve people’s livelihood and facilitate sustainable practices (https://sdgs.un.org/, accessed on 20 June 2024), and it is crucial to preserve global biodiversity and bridge the gap between the microbiome and its role in global health [16] (Figure 1).

The rapid developments in fungal biology have facilitated the development of biomass-conversion technologies, and the production of high-value substances as food and feed components. Microbe-based sustainable innovations would likely contribute to United Nations SDGs by providing affordable energy (use of agro-industrial waste by microbial conversions), reducing economic burdens/affordable living conditions (new opportunities by the creation of bio-based industries for sustainable living), tackling climatic changes (use of sustainable alternatives fuels for reducing carbon footprints), conserving marine life (production of microbe-based bioplastics for safer marine life) and poverty reduction (microbial products/microalgae farming), among other initiatives [17,18,19]. These objectives can be achieved via fungal biotechnologies to enhance the production of metabolites, chemicals, and proteins, microbial processing (using microbial enzymes), and advances in biorefineries to develop high-value products. Field and coworkers [20] discussed the potential of mycorrhizal associations as a sustainable approach to achieving food security, conservation, and SDGs [20]. It is important to mention that many mycorrhizal associations of fungi form edible mushrooms, while their collection and consumption are significant for nutrition, traditions, and the global economy [21]. Furthermore, staple cereal crops [22,23,24] and high-value food crops (vanilla flavors) [25] benefit from mycorrhizas, with an important yet overlooked impact on human societies and the ecosystem.

Delving into how the advances in fungal biotechnologies can attain SDGs, state-of-the-art concepts, transformative approaches, achievements, and prospects/directions in the future are discussed in this paper.

## 2. Fungal Biotechnologies and SDGs—How Far We Have Come

In the face of climate adversities and changing landscapes, human reliance on fossil fuels has impacted productivity and lifestyles and has driven increased emission rates and environmental deterioration [26]. The increased recognition and need to prioritize sustainable practices [27,28] to address and regulate the environmental impact of human activities [29] have been the main goals of SDGs of the United Nations.

The enriched yet less tapped fungal biodiversity can contribute to achieving SDGs, a prospective initiative of the United Nations [30]. Fungal species provide transformative opportunities from petroleum-based to bio-based economy opportunities attributed to converting organic substances into diverse ‘high-value’ products for addressing socio-economic concerns. The utilization of fungal bio-based products is sustainable in securing and enhancing the food supply for a growing population and limiting greenhouse emissions. In addition, the advances in fungal biotechnologies have the potential to tackle global climate change and accomplish SDG reforms (Table 1).

### 2.1. Fungal ‘High-Value’ Products to Achieve Global Food Security, Tackle Hunger and Malnutrition

Unlocking the road to sustainable food production is challenged by the growing world population, climate fluctuations, food prices, global catastrophes, and agricultural losses due to pathogens [82,83,84]. The development of bio-based products via fungal biotechnologies demonstrates potential in reducing hunger and malnutrition and ensuring food security. Moreover, multiple lifestyle diseases can be tackled by functional foods and nutraceuticals of fungal origin [85,86], following balanced nutrition. Alternative food resources have gained key consensus due to their beneficial health impact and nutritional value. The multi-faceted aspects of food components are improved following microbial synthesis including bio-functionality, quality/nutritional value, peptide synthesis, antimicrobial function, and reduction in antinutritive components, etc. [87,88]. Fungi-based food demonstrates potential as a high-nutritional source for addressing global hunger and malnutrition, besides demonstrating industrial importance (Figure 2).

Among the 2–11 million fungal species in nature, only a fraction (approx. 1.5 lakh species) have been reported; furthermore, only some adhere to the acceptable guidelines of functional food. Since prehistoric times, fungal species have been used to prepare beverages and food products including cheese, bread, food flavors, etc. Only recently, the horizon has expanded to other biotechnological utilities. The prospective pharmaceutical/industrial use of economically viable strains can be attributed to the tractability and transformative potential of fungi, which lead to horizontal gene acquisition and overall plasticity [89]. The edible mushrooms from phyla *Ascomycota* and *Basidiomycota* are widely utilized in food preparations across the globe and viewed as exquisite delicacies. The notable fungi namely *Aspergillus*, *Penicillium*, and *Fusarium* sp. are widely recognized for their nutraceutical/pharmacological properties. The fungal mycelium comprises dietary fiber, health-promoting lipids, and vitamins and has health benefits. Some edible fungal species are key sources of probiotics and food flavors [90], while certain filamentous fungi are good protein sources (high protein content) [91]. The food derived from fungal sources has the following major advantages: amino acid profiles, high nutritional and protein content [91,92], and high concentration of fibers, vitamins, and unsaturated fats in the case of edible mushrooms [93]. Research into harnessing the socio-economic benefits of fungi has delved into developing food components comprising nutraceuticals, functional food products, pharmaceuticals, and enzymes [9,94]. Key studies have documented the bioactivities of fungal constituents. Polysaccharides from *Morchella esculenta* promote antioxidant enzyme function [33], *Ganoderma* enhance immune functions [95], *Tremella* relieves epidermal bleeding [96] and *Agaricus bisporus* restricts the growth of cancer cells [97], among other examples. Health promotion effects are demonstrated by oligosaccharides from copropilous fungi and are developed as a type of functional food [98]. With the advent of white fungal biotechnology, the quality and nutritional value of food products have remarkably improved in the flavor of bread and beverages, single-cell protein (SCP) quality and the yield and shelf life of products [99]. Worldwide, mushrooms are considered to be major aspects of various cuisines and highly nutritional sources of carbohydrates (60%), protein (27–48%), and lipids (2–8%) [100], amino acids (glutamine, valine, leucine, etc.) and vitamins [101,102]. The commercially cultivated mushroom species are represented by *Agaricus bisporus*, *Pleurotus* spp., *Auricula auricula*, *Lentinus edodes*, and *Volvariella volvacea* [103,104]. The commercial market for mushrooms has witnessed a tremendous upsurge, with the value for oysters, shiitake, and champignons exceeding USD 50 billion by 2022 [105].

Tian and coworkers [106] showed the beneficial effects of *A. bisporus* on glucose homeostasis, and the prebiotic effect on glucose homeostasis and regulation of diabetes. In C57BL/6 mice, succinate and propionate produced by *Prevotella* sp. signals intestinal gluconeogenesis, affects the gut–brain neural circuit, and reduces glucose in hepatic cells. The growing awareness about the nutritional components in multiple fungal species and their increased consumption has raised the demand, and sustainable methods are being employed to meet the increased demands globally. The development of novel strains via genetic engineering studies would be a prospective approach to increase the desired product yield and productivity. Fungal species are ideal resources to develop alternate food components, novel drug molecules, and maintain environmental sustainability.

### 2.2. Harnessing Pharmaceutical Metabolites from Fungi in Healthcare

SDGs established by the United Nations aim to attain sustainable growth and holistic upliftment of human lives by 2030, utilize alternative bio-based resources, and address global issues. The transition from a fossil-based to a bio-based economy requires the integration of advanced biotechnologies with bioeconomy [107]. The diverse yet interesting group of known fungi inhabits different ecological niches and contributes to multi-faceted roles in the environment, ranging from symbionts, and decomposers to pathogens.

Fungal species produce a plethora of diverse, ‘high-value’ compounds including therapeutics, food components, biofuels, chemicals, vegan leather, organic acids, industrial materials, etc., that can be effectively utilized for sustainable living. Fungi, as the major drivers of bio-based economy, demonstrate diverse fermentation capabilities (industrial value) attributed to their active metabolism (ecological relevance) and adaptation to wider niches (industrial applications). The commercially important high-value products, namely antibiotics and drugs, can be utilized to treat human ailments and positively impact human health and well-being. To date, thousands of pharmacologically active metabolites have been purified and characterized using fungi- demonstrating potent efficacies in treating multiple disorders [108,109].

The landmark discoveries of penicillin and cephalosporin C from fungi opened new avenues and revolutionized fungi-mediated drug discovery. Constituting both classes of traditional drugs and recent landmarks, fungi-derived drugs have been effective in treating the following chronic diseases: autoimmune disorders (immunosuppressants), hypercholesterolemia (statins), and chronic infections (antifungal and antibiotics) [110]. The representative examples include cephalosporins (antibiotic), penicillin V (antibiotic), fusidic acid (antibiotic), griseofulvin (antifungal), retapamulin and enfumafungin (antifungal), among other notable examples. The translational success of these drugs can be attributed to their validation in clinical trials (drugs for drug-resistant depression and cancer). Subsequently, fungal-derived immunosuppressants, such as cyclosporin A (from *Tolypocladium inflatum*), block the calcineurin pathway (hampering T-cell activation in humans) and have been pivotal in organ transplantations [111]. Another drug (isolated from *Penicillium brevicompactum*) named mycophenolic acid hampers inosine monophosphate dehydrogenase and biosynthesis of guanine, which restricts the proliferation of lymphocyte (in organ transplantations) [112]. The synthetic compound, fingolimod (inspired by fungi-derived myriocin), is produced by Novartis and achieved blockbuster success as an immunosuppressant for multiple sclerosis, generating USD 1 billion in 2012 [113]. Tiwari et al. [2] extensively discussed and highlighted the potential of plant-associated endophytes to produce potent antimicrobials and counter drug-resistant microbes, an emerging medical concern in the present era. The antimicrobials, namely hypericin, cryptocandin, leucinostatin A, colletotric acid, munumbicins, and their derivatives demonstrated clinical efficacies in treating drug-resistant pathogens; however, assessment and further trials are imperative to establish their therapeutic potential and drug development. In obstetric medicine, bromocriptine (a synthetic form of ergocryptine), is a dopamine agonist and restricts prolactin release from the pituitary gland [114,115]. It is used for the treatment of hyperprolactinemia-related conditions. In therapeutic advances for treating blood cholesterol levels, fungal-derived drugs have proved pivotal in achieving key success. The discovery of mevastatin (compactin) from *Penicillium citrinum* by Akira Endo, a Japanese scientist, ushered in a new era [34]. Lovastatin (the statin drug), isolated from *Monascus ruber* (documented as monacolin K) [116] and subsequently from *Aspergillus terreus* (documented as mevinolin) [117], was quite successful in lowering blood cholesterol. Lovastatin was successfully marketed as a cholesterol-lowering drug in 1987, followed by mevastatin [118]. Statins comprise one of the highest-marketed drugs worldwide, generating sales of USD 25 billion in 2005. Furthermore, several compounds of fungal origin and their derivatives are currently in clinical trials for multiple diseases and include Halimide (synthetic derivative Plinabulin) in phase III trials for cancer, Hypothemycin (synthetic derivative E6201) in phase I trials for solid tumors/melanoma, Wortmannin (synthetic derivative PX-866) in phase II clinical trials for prostate cancer, Cordycepin (synthetic derivative NUC-7738) in phase I trials for lymphoma/solid tumors and Radicicol (synthetic derivative Ganetespib) in phase III trials for lung cancer, among other therapeutics. Gomes and coworkers [93] have extensively discussed and highlighted the importance of marine-derived fungal metabolites for cancer treatment, including leptosins, gliotoxin, shearinine, meleagrin, neoechinulin A, and bostrycin, etc. [119]. The biosynthetic gene clusters (BGCs) in fungal genomes synthesize bioactive, high-value metabolites and can be investigated/engineered for obtaining higher yields of the targeted metabolites.

### 2.3. Novel Fungal Cell Factories for the Production of Bioactive Metabolites

Fungal-derived metabolites exhibit enormous diversity and interesting bioactivities, namely antimicrobial, hypoglycemic, antiviral, antitumor, immunosuppressant bioactivities, etc. The increasing evidence from studies highlights the potent efficacies of fungal-derived bioactive metabolites as key therapeutics. In addition, functional food/nutraceuticals from fungi have been documented to promote human health and well-being [9,120] and multiple fungal species are powerful resources used to generate ‘high-value’ substances of socio-economic relevance.

Filamentous fungi are widely recognized as efficient producers of natural products, industrial substances, enzymes, proteins, organic acids, etc., and are employed as novel tools for targeted morphology engineering [121]. In addition, fungal biomass is also important in textile industries and as a food component. Discussing industrial relevance, fungi produce key enzymes including phytases, proteases, catalases, and glucoamylases and others with wider usage [122]. Fungal enzymes are also utilized in biofuel production to convert lignocellulosic biomass to fermentable sugars, generating an economic return of over EUR 4 billion [8]. For large-scale cultivation (both solid-state and submerged fermentations), understanding and reprogramming fungal morphogenesis and growth are crucial. Further efforts are needed in process design to optimize fungal morphology for producing a targeted product. Multiple investigations/research in this direction have speculated that septal secretion in fungi may have industrial value and optimization of fungal morphology would improve septal junctions by genetic manipulation studies, in addition to prospective yet less-explored intercalary secretion pathways [123].

Fungal secondary metabolism and its exploration are promising, with studies suggesting that more than 60% of medicines comprise natural products [124]. While efforts are being made for the bio-prospection of fungal resources, new techniques for the activation of silent gene clusters (BGCs) in the laboratory and pilot fermentation studies [125] have been employed and enhanced production via targeted genome manipulation has been achieved [126]. The advances in synthetic biology and a deeper elucidation of the filamentous life cycle for fungal genome engineering facilitate targeted strain development [123].

*Aspergillus* is fast emerging as a model for genome manipulation, attributed to the technological advances in whole genome sequencing. Engineering initiatives started in the 1950s, ranging from manipulating fungal morphologies and mutagenesis to achieving high product titers. For instance, strains of *A. oryzae* were subjected to nitrous acid and UV mutagenesis, resulting in less viscous broth and higher production of glucoamylase [127]. Subsequently, mutagenesis of *Trichoderma reesei* with diethyl sulfite led to a highly branched and short chimeric strain showing enhanced cellulase production [128]. Through the efforts in genome sequencing, an increased understanding of the candidate genes/metabolic pathway has been achieved for strain improvement [14]. In addition, attempts have been made for single nucleotide polymorphism (SNP) identification in fungal genomes for better growth of fungal strains; however, studies are limited. The signaling pathways govern morphological regulation in fungi, and engineering attempts have been made to target components in the cascade for enhanced biotechnological utilities. The key signaling pathways in filamentous fungi, protein kinase A (PKA)/cyclic adenosine monophosphate (cAMP) signaling, calcium ion responses, and mitogen-activated protein kinase (MAPK) are the prime focus of targeted fungal engineering for fungal growth and morphological improvements [123].

In this direction, synthetic biology has made significant advances to create designer chimeras possessing minimized genomes, less complexity, and improved attributes, respectively. A reduced genome of *S. cerevisiae* was created, and a significant portion (14%) of chromosome 3 was deleted (tRNA, transposans, and wild-type base pairs) [129]. The genome editing of *S. cerevisiae* chromosome 16 and fusion experiments resulted in chimeras with reduced genome size [130,131]. In *A. niger*, targeting the fungal genome for minimization was achieved by deletion/inactivation of certain genes/chromosome sections by the CRISPR-Cas9 tool [132]. The genomes of economically viable fungal species have been engineered by the CRISPR-Cas9 editing system and are as follows: available *T. reesei* [133], *M. thermophila* [134], *A. oryzae* [135] and *P. chrysogenum* [136], among other notable ones, and are exploited industrially. The concept of engineering fungal genomes for size reduction relies on the deletion of non-essential genes and focuses on targeted genes/pathways for morphological changes/growth phenotype and the creation of desired fungal chimeras.

The natural product discovery pipeline has been greatly expedited, which is attributed to the advances in the synthetic biology toolkit. In addition to CRISPR-Cas, advanced synthetic tools are promising in creating designer fungal cell factories, improving the morphological feature and high titer of the desired metabolite. The heterologous expression of key biosynthetic genes (for natural products) has been achieved in *A. niger* [38], *A. nidulans* [137], and *P. chrysogenum* [138], etc., and synthetic fungal chimeras with new/novel attributes have been created by domain swapping [139] and fungal media optimization. Synthetic biology toolkits have expanded the horizons to facilitate polycistronic gene expression in filamentous fungi [140], and next-stage morphological engineering via controlled gene expression of multiple genes using a single promoter, offers interesting insights [141]. Successful attempts in engineering and optimization of tuneable gene switches in filamentous fungi [142] offer precise details of the strain’s morphological characteristics and gene function. It is imperative that advances in fungal imaging have provided precise information about fungal morphology; X-ray microtomography has elucidated the three-dimensional morphology of *P. chrysogenum* and *A. niger* [143] and defined new prospects in precise quantification of hyphal number distribution in the pellet, providing future directions in understanding how pellet morphologies affect the titer of the product. These technological developments, optimized in filamentous fungi and other fungal systems (in progress), will result in novel fungal cell factories, including minimizing genomes, higher product titers, and optimized fungal morphologies, in the future.

### 2.4. Fungi-Based Bioremediation for Environmental Subsistence

The present era has witnessed an increased interest in microbial biodegradation of toxic contaminants for ecosystem restoration. Microbe-assisted bioremediation comprises a cost-effective and eco-friendly approach for the transformation of recalcitrant pollutants into environmentally degradable substances. In addition to other microbial species, fungi-mediated bioremediation is a safe and renewable strategy for mitigating contaminants/polluted locations [144]. Fungi play a critical role as degraders and symbionts, colonizing diverse environmental niches and possessing consistent morphology and multi-faceted metabolic potential. A combination of biostimulation, bioaugmentation, natural attenuation, or individual approaches can be used [145] as per the requirement and efficiency of the microbial strain. Mycoremediation has been a method of choice in environmental cleanups, with multiple efficient fungal species documented for their potency in mitigating heavy metal contamination, pollutants, greenhouse gases, industrial chemicals, etc. [146,147].

Mycorrhizal fungi play a key role in the ecosystem by promoting plant access to nutrients and water in soil and plant tolerance to pathogens. In addition, fungal species in mycorrhizal associations contribute to bioremediation, conservation, and ecosystem well-being [148]. Mycorrhizal associations confer salt tolerance to the plant and promote plant growth and overall health. Bioremediation achieved in the capacities of microbial degradations minimizes the amount and harmful impact of diverse contaminants, while microbial processes aid in the mitigation of pollutants in contaminated sites. Microbe-assisted chemical and physical processes cause disintegration and structural changes in the pollutants and accelerate metabolism. In addition, microbes facilitate energy-dependent chemical reactions for the dissemination of contaminants and electron transfer [149] via oxidation and reduction reactions. In nature, microbes acquire carbon from contaminants for growth and degrade them into simple substances. Quite interestingly, mycoremediation is effective in the removal of heavy metals and radioactive agents to be further decomposed [150]. During pesticide degradation, fungal species obtain nitrogen, carbon, or energy for growth. Molds, e.g., *Botrytis* and *Aspergillus,* decompose sugar polymers, celluloses, starches, pectins, oils, chitin, oil components, etc. Subsequently, an environmental hazard, e.g., endosulphan, is effectively degraded by *Trichoderma harzianum*, *Cladosporium oxysporum*, *Aspergillus* spp., and *Mucor thermohyalospora*. Moreover, fungi-mediated degradation of pesticides into non-toxic substances occurs via processes namely hydroxylation, dehydrogenation, esterification, and deoxygenation [151]. Other fungal strains are capable of bioremediating different contaminants, including textile wastewater detoxification by *Zygomycetes* and *Aspergillus*, polychlorinated biphenyl (PCB) degradation by *Fusarium solani*, *Penicillum chrysogenum* and *Penicillum digitatum*, biosorption of pentachlorophenol by *Rhizopus oryzae* CDBB-H-1877, cellulose degradation by brown rot fungi, xenobiotics degradation by *Agaricus bisporus*, *Pleurotus ostreatus*, *Pleurotus pulmonarius*, etc., heptachlor and heptachlor epoxide remediation by *Phanerochaete ostreatus* and effluents from textile industries by *Ascomycetes* and *Basidiomycetes* fungi, among other distinct examples [151].

Besides the remediation of contaminants present in the environment, the restoration of polluted sites has been achieved via naturally occurring microbes. The representative examples *Penicillium*, and *Aspergillus* were effective alleviators of contaminants like textile dyes, chemicals, pesticides, industrial effluents, organic pollutants, etc. [152,153]. In addition, the substantial removal of petroleum hydrocarbons and diesel contaminants in soil has been successful by short-term inoculation of *Phanerochaete chrysosporium* and *Aspergillus niger*, which facilitated bioremediation [154,155]. Literature studies have shown that white rot fungi disintegrate harmful substances, namely phenols, effluent, pesticides, heavy metals, polychlorinated biphenyls, etc., and alleviate the adverse impact on soil. Studies have also established the significant potential of fungal enzymes (lipases, catalases, amylases, proteases, peroxidases, etc.) in organic waste management [156], highlighting their industrial value [157,158]. Advanced technologies have immensely contributed to addressing limitations with fungi-mediated bioremediation. In recent times, immobilized fungi in bioreactors (fluidized bed reactors and rotating biological reactors) have been adopted for bioremediation [159,160]. For the treatment of wastewater sludge from sewage plants, it is mixed with microbial inoculum in a broad-scale bioreactor and considered a sustainable approach [161,162]. Furthermore, advanced practices for PAH mitigation include *Trichoderma longibrachiatum*-based biobarriers on nylon sponges, where high efficiency of PAH removal was achieved [163]. An upcoming approach utilizes yeast expression systems to generate cytochrome P450 monooxygenases that can tackle hydrocarbons and aid in mitigation [164].

### 2.5. Addressing Climate Changes via Fungal Biotechnologies

For addressing climate change, it is imperative to achieve net-zero emissions by the mid-century to limit temperature rise within 1.5 °C, while adopting measures to sequester, capture, and store excess atmospheric carbon [165]. In a recent report by the World Economic Forum, fungi can play a crucial role in addressing climate change [166]. Fungal species inhabiting natural environments assist forests in absorbing carbon and tackling the potential impacts of climatic fluctuations. While fungi occupy diverse ecological niches and mushrooms are present in shady and damp places, mycorrhizal fungi (ectomycorrhizal fungi) assist trees and forests to absorb CO_2_ faster and reduce the rate of carbon flow/return in the atmosphere. However, the rapid deforestation every year threatens the beneficial interactions, and promoting the regrowth of forests would reduce global emissions by 30%, as per the guidelines of the COP26 summit in Glasgow [166]. Since little information is available on the role of fungal networks in combating climate change, the Society for the Protection of Underground Networks (SPUN) has devised a project to understand the role of mycorrhizal fungi in areas of climate science and map the ‘Wood Wide Web’. Thousands of fungal samples are collected to map fungal networks and utilized by SPUN (via machine learning) to create these networks and their function as carbon sinks. This information could be used to identify high-priority zones for more carbon storage and tackle extreme conditions.

The growing evidence from the literature suggests that fungi can contribute to farming practices and agriculture. The inoculation of seeding soil with beneficial fungi promotes soil attributes, enriches soil fertility, and decreases atmospheric CO_2_ levels, crucial to environmental functioning [167,168], while pathogenic macrofungi exploit plants and animals to absorb nutrients and also contribute to biodiversity in the ecosystems. The interconnected mycelial network with the host is crucial and improves nutrient acquisition, transport, and enzyme secretion [169]. These fungi-mediated processes are essential for sustaining biodiversity and ecosystem well-being. The translational success of arbuscular mycorrhizal fungi (AMF) as potential biofertilizers has a major impact on the global market with a value of USD 2 billion. In addition, fungi are key players and perform essential functions in the ecosystem. Globally distributed, fungi carry out processes including bioconversion, energy flow processes, and nutrient cycling and act as symbionts, pathogens, and decomposers in nature [170,171]. According to a study in nature, the biodiversity of fungi determines plant biodiversity, productivity, and variation in the ecosystem and approximately 90% of plants form integral associations with fungi. In addition to other functions, fungi perform mycoremediation (as earlier discussed), degrade chemicals heavy metals, crude oils, etc., absorb heavy metals and radioactive components, and maintain ecological subsistence. However, excessive human activities and pollution levels are impacting fungal diversity/population and signaling climate change. Adequate and urgent efforts are required to stop/minimize deforestation, restore ectomycorrhizal forests, and switch from fossil fuel to renewable energy sources (as in America).

## 3. Achievements and Prospects in the Present Decade: What Do We Know and What Comes Next?

Recently, the United Nations General Assembly (UNGA) Science Summit stated that “understanding the world of microbes is imperative either to curb dangerous effects or to harness their power for healthier life, for sustainable energy sources, for biodiversity, for tackling climate change and for solving hunger problems”, which is one of the key objectives of the United Nations SDGs. The microbes in the environment are integrally associated and impact human lives. The increased recognition of the favorable impact of beneficial microbes on humans and the environment has contributed to their potential applications in healthcare, agriculture, and ecosystem restoration.

Widely exploited as a source of ‘high-value’ food ingredients (food flavors, pigments, nutritional substances, etc.), the present era has witnessed the utilization of fungi-based biofertilizers to boost crop health and productivity. Moreover, water quality and sanitation have been remarkably improved by microbe-assisted remediation of contaminated water bodies. Other achievements in microbial biotechnologies in achieving SDGs have been biofuel production as a direct source of affordable and clean energy, industrial production of high-value metabolites, potent drugs from fungi approved and marketed for disease treatment with others in different stages of drug development, environmental cleanup via bioremediation of contaminants and plastics and conferring stress tolerance to plants in times of global climate adversities.

Cutting-edge research has focused on deciphering and highlighting the prospects of beneficial microbes in different socio-economic contexts. With the beginning of the transition towards a bio-based economy and the efficient utilization of fungal resources, answers to the following pertinent questions are required: which species has valuable/useful traits, and how can self-sustainability be achieved by fungal production [1]? A better understanding can be achieved with these answers on the road towards a sustainable future. In light of the current findings, it is important to investigate/screen the vast repertoire of fungal species and validate the bioactivities, which are necessary to define the safety profiles for socio-economic applications.

Advanced biotechnological tools have revolutionized the exploration of natural resources. The phylogenomics-guided exploration of specific traits has been inferred from the relationship between microbial species. In addition, progress in analytical equipment and omics-assisted identification of species have contributed to bridging the knowledge gaps in metabolite biosynthesis and evaluation of their bioactive potential [12,172]. Metabolomics studies have attempted to understand the fungal metabolic networks and their dynamics, providing critical insights into the taxonomic identification, fungal stress response, metabolite discovery, metabolic engineering, and plant–fungal interactions. A deeper knowledge of complex fungal interactions and their environmental responses has been attained via metabolomics [172,173]. Omics biology has also contributed to research on edible fungi (cointegrated with other methods), delving into processes including stress resistance, growth and development, and its pharmaceutical value [174,175], providing in-depth information. With the advent of modern genome editing tools, like CRISPR-Cas, the production of high-value metabolites can be optimized by fungal genome engineering, heterologous expression, and gene disruption [13], among others. Molecular analysis of the fungal genome provides a framework to screen beneficial traits, metabolite discovery, and efficient production under laboratory conditions. The key to strengthening fungal resources and biotechnologies to achieve the sustainable goals involves obtaining extensive knowledge of fungal biology (a global effort is needed), building a global network, and providing a knowledge base platform for fungal identification, classification and collection of fungal species [176].

## 4. The Road Ahead: Future Directions in a Fungal Bio-Based Economy

The enriched yet less tapped fungal biodiversity can contribute to realizing SDGs, a prospective initiative of the United Nations for a better world. Fungal biology and biotechnologies provide transformative opportunities from petroleum-based to bio-based economies attributed to converting organic substances into diverse ‘high-value’ products for socio-economic sustainability. Fungi have been associated with land plants during their evolutionary course, and harnessing the power of ancient players would benefit natural habitats and biodiversity [20]. The utilization of fungal bio-based products is sustainable in securing and enhancing the food supply for a growing population and limiting greenhouse emissions. The development of alternate food products includes Quorn (meat substitute) (https://mycorena.com, accessed on 20 June 2024) [177], filamentous fungi-based biomaterials [178], biorefinery applications (second-generation biofuels) [179], biodegradation of plastics [180], and other notable examples. In addition, the advances in fungal biotechnologies have the potential to tackle climate change and contribute to the United Nations SDGs. The road to sustainable development is not yet reachable, and fungal resources represent prime resources in addressing sustainable livelihood and development in a global context.

## Figures and Tables

**Figure 1 jof-10-00506-f001:**
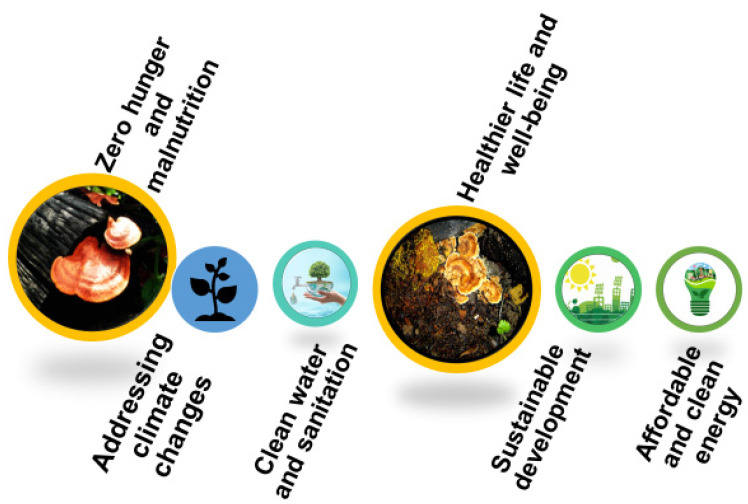
Sustainable development goals (SDGs) and role of fungal biotechnologies.

**Figure 2 jof-10-00506-f002:**
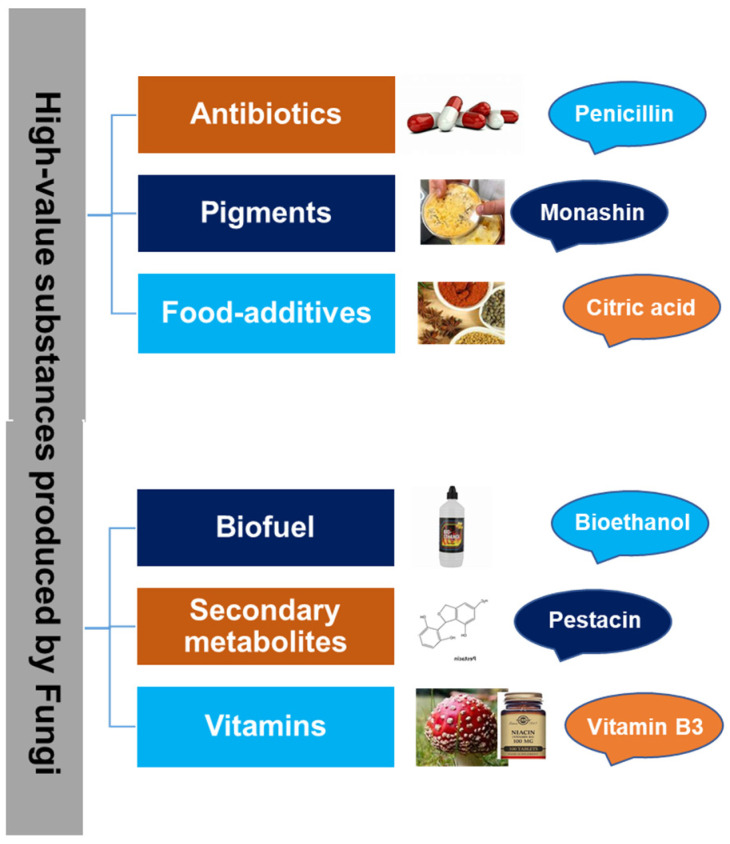
Development of a bio-based economy via production and utilization of high-value products from fungi [9].

**Table 1 jof-10-00506-t001:** Representative examples of ‘high-value’ products from fungi (natural and engineered strains) and their potential to achieve SDGs.

Fungal Species	High-Value Product	Biotechnological/ Economic Utilities	References
Fungal high-value metabolites in medicinal applications			
*Acremonium chrysogenum*	β-lactam antibiotics (cephalosporins)	Pharmaceutical value	[31]
*Lentiana edodes*	Lentinan	As chemotherapy adjuvant in healthcare	[32]
*Penicillium rubens* *P. solitum* *P. chrysogenum*	Penicillin Mevastatin β-lactam antibiotics (penicillins)	Pharmaceutical value Statins are widely used in lowering blood cholesterol levels Antibiotics in healthcare	[33,34,35]
*Saccharomyces boulardii*	Probiotics	Health supplements	[36]
*Aspergillus terreus* *A. niger*	Secondary metabolites (lovastatin) Secondary metabolites (enniatins) Human granulocyte colony-stimulating factor (G-CSF) Galactaric acid	Pharmaceutical value Enhanced production of high-value metabolites High protein titre for medicinal applications Efficiently produce galactaric acid for industrial applications	[37,38,39,40]
*Rhizoctonia bataticola*	Forskolin	Anti-HIV, anti-tumor, therapeutic application	[41]
*Phomopsis* sp.	Quinine	Antimalarial, used in malaria treatment	[42]
*Alternaria* sp.	Digoxin	Cardiotonic, therapeutic application	[43]
Fungal species in food industries/food applications			
*Blakeslea trispora*	Carotene Lycopene	Food pigments for application in food sector	[44,45]
*Monascus anka*	Monascus pigments	Food pigments as natural food colorants	[46]
*S. cerevisiae*	Lycopene (carotenoid) Ethanol Production of fatty acid-derived biofuels Terpene production	Food pigment for use in food sector Biofuel production Industrial applications Genetic engineering for enhanced terpene production	[47,48,49,50]
*Morchella esculenta*	Polysaccharides	As food (nutritional) supplement	[51]
*Pleaurotus eryngii*	Pork sausage (food component)	Used as food component	[52]
*Fusarium venenatum* *Fusarium* sp.	Quorn (meat substitute) Dairy-free cream cheese	Nutritional food (high amino acid and fiber, fungal protein) Food industries	[53,54]
*Penicillium camemberti* *P. roquefortii*	Production of cheese Blue cheese	Food industries ---	[55,56]
*Mushroom mycelium*	Plant-based bacon	Alternative food product	[57]
*Aspergillus* sp. *A. oryzae* *A. sojae* *A. niger*	Fermented meat Soy sauce Miso Jiuqu Citric acid Enzymes	Alternative meat source, high protein content Traditional fermented food --- Food industries	[58,59,60,61]
*Yarrowia lipolytica*	β-carotenoid	High metabolite yield for food sector application	[62]
*Xanthophyllomyces dendrorhous*	Zeaxanthin	Food pigment usage in food industry	[63]
*Mortierella alpina*	Linoleic and oleic acids	Food industry	[64]
*L. edodes*	Pasta (functional food)	Nutritional supplements	[65]
Fungal metabolites for industrial applications			
*Ustilago maydis*	Itaconic acid	Bio-based building block in the polymer industry, pharmaceutical value	[66]
*Kluyveromyces lactis*	L-ascorbic acid (vitamin C)	Enhanced production for industrial applications	[67]
*Trichoderma reesei*	Enzyme (cellulase)	Enhanced production for industrial applications	[68]
*Schizophyllum commune*	Textiles	Industrial application	[69]
*Ganoderma lucidum*	Composite material, construction material	Biomaterials to reduce environmental pollution	[70]
*Umbelopsis isabellina*	Constituents of biodiesel (polyunstaturated fatty acids)	Biofuel production, energy source	[71]
Fungal metabolites for agricultural applications			
*A. nidulans*	Insecticides (austinoids)	Production of austenoid derivatives including 7-hydroxydehydroaustin, 1,2-dihydro-7-hydroxydehydroaustin, 1,2-dehydro-precalidodehydroaustin, calidodehydroaustin, etc.	[72]
*Beauveria bassiana*	Mycoinsecticides	Integrated pest management, biocontrol of arthropod pests	[73,74]
*Trichoderma* spp. *T. harzianum* T22 *T. harzianum* TC39	Auxin-like metabolites, proteinaceous compounds Azaphilone, harzianolide, 1-hydroxy-3-methylanthraquinone and harzianopyridone	Regulate plant growth and development, agricultural applications Biocontrol agents, suppress the growth of plant pathogens	[75,76,77]
*Gliocladium virens*	Antifungal compounds gliovirin, viridiol, valinotrocin, viridin, gliotoxin, and heptelidic acid	Protect agricutural crops from multiple pathogens, bicontrol functions	[78]
*Botrytis cinerea*	Abscisic acid	Phytohormone regulates abiotic stresses, application in agriculture	[79]
*Chaetomium globosum* Cg-7, *C. globosum* Cg-6 *C. globosum* Cg-5	Chaetoglobosin	Reduce post-harvest diseases in multiple fruits	[80]
*Eupenicillium parvum*	Azadirachtin A and B	For the control of insects, biocontrol functions	[81]

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
