# Peer review of "Advanced Fungal Biotechnologies in Accomplishing Sustainable Development Goals (SDGs): What Do We Know and What Comes Next?"

_jof, 2024, doi:10.3390/jof10070506_

Round 1

Reviewer 1 Report

Major issues:

1.     The title may need revising. To the best of our knowledge, “accomplish (achieve) xxx goal”, rather than “address xxx goal”.

2.     Wording redundancy for example, in the Abstract section (P1 Line 25-27) there is a paragraph as follows, “Delving into how the advances in fungal biotechnology can contribute to/address sustainable development goals (SDGs), state-of-the-art concepts, emerging trends, achievements and prospects/directions in the future, are discussed”, however, at the end of the Introduction section, P3 line 112-114, the author wrote “Delving into how the advances in fungal biotechnology can contribute to sustainable development goals (SDGs), state-of-the-art concepts, transformative approaches, achievements, and prospects/directions in the future, are discussed.

There are  a number of repetitious expressions in every sections throughout this manuscript. For instance, in my opinion, the first paragraph and 2nd paragraph of the Introduction section may be merged to one.

3.     Table 1 Representative examples of high-value products from fungi (natural and engineered strains) and their potential to achieve Sustainable Development Goals (SDGs). As for S. cerevisiae, there are three places to exemplify its use in lycopene (carotenoid), ethanol, and fatty acids-derived biofuels, respectively. May these 3 cases be merged into one, under the fungal species “S. cerevisiae”? Same things for A. niger.

4.     P 6, Figure 2Development of a bio-based economy via production and utilization of high-value products from Fungi: The Fig. 2 in this manuscript is totally same as the Figure 1 published in Microorganisms 2023, 11, 1141 by Tiwari and Dufosse. A  screenshot of Fig. 1 by Tiwari and Dufosse (2023) is attached below.

To the best of mu knowledge, the admission from the original authors (Tiwari and Dufosse, 2023) /or from the publication journal should be authorized depending on the relevant copyright regulations.

Minor issues:

1.     In the Abstract, P1 line 19, when the abbreviation SDGs was initially mentioned, it should be described in full like sustainable development goals (SDGs).

2.     P1 line 32, depleting > exhausting

3.     P1 line 39,  With significant association and influence on human lives > with significant association with and influence on human lives

4.     P2 line 46-47, expanded for commercial use in food components, the chemical and pharmaceutical sector > expanded to commercial use in food additives, the chemical and pharmaceutical sectors.

5.     P2 line 48, to remove ‘being’

6.     Please rewrite the paragraph of P3 line 91-93: “In the last decade, the rapid expansion in fungal biology translated into cascading biomass-conversion technologies, unlocking full potential and facilitating the production of high-value substances as food and feed components.

7.     P3 line 112-113, to remove the full term for SDGs since it has been defined previously.

8.     P4 line 122, realizing > achieving

9.     P4 line 128-129, to tackle climate change > to tackle global climate change

10.  P6 line 136-138, please rewrite the sentence starting with “Fungal biotechnology and bio-based products”. I would suggest the authors decrease the wording/expression redundancy.

11.  P6 line 138-139, managed > tackled

        P6 Line 140, achieved key recognition  > gained key consensus

        P6 line 153, time immemorial > prehistory time;

        P6 line 153-154,  food products should include cheese, bread, food flavors, etc.

12.  P6 line 157-159, in the sentence of “Discussing bio-based components, edible mushrooms from phyla Ascomycota and Basidiomycota are widely utilized in food preparations across the globe.”. Do the authors express that the “discussing bio-based components, edible mushrooms…are utilized in the global food preparations” or  “bio-based components, edible mushrooms…. are widely utilized in food preparations across the globe.”

  I would suggest the word ‘discussing’ should be removed.

13.  P7 line 168-172, Please rewrite the sentence starting with “Key studies have documented…”

14.  P7 line 189, to achieve > to meet

15.  P7 line 190, method > approach

16.  P7 line 192, new/novel–> novel

17.  P7 Line 192-193, achieve > maintain

18.  P7 line 199-200, “in the environment from symbionts, and decomposers to pathogens>”in the environment ranging from symbionts, decomposers to pathogens”

19.  P7 line 202,  …, deal with > …, to deal with;

P7  line 203,  and tackle >  and to tackle

20.  P9 line 316-318,  The concept of engineering fungal genomes for size reduction relies on the deletion of non-essential genes, and focus on targeted genes/pathways for morphological changes/growth phenotype and the creation of desired fungal chimeras.> The concept of engineering fungal genomes for size reduction relies on the deletion of non-essential genes, and focusses on targeted genes/pathways for morphological changes/growth phenotype and the creation of desired fungal chimeras.

21.  P10 line 324, there should be a coordinating conjunction like ‘and’ to link the two sentence.

22.  P10 line 327, remove the repetitive phrase “in filamentous fungi”

23.  P10 line 336-339, please change the phrase ‘contribute to’ to another one, such as bring about, result in, lead to, pave the road towards…. Throughout the manuscript,  the phrase of ‘contribute to’ had been used for 14 times.

24.  P10 line 351-352,  potent role > potency

25.  P10 line 354-356,  expression redundancy for “…have great potential and play a key role in…”

26.  P10 line 374-382, bioremediation > bioremediating;

textile wastewater can be detoxified by…> textile wastewater detoxification by…

27.  P10 line 384-387, please rewrite the sentence starting with “Fungi classified into…”

28.  P10  Line 399-401, “advanced approaches for PAH mitigation include Trichoderma longibrachiatum-based biobarriers on nylon sponges, high efficiency of PAH removal was achieved” …> advanced approaches for PAH mitigation include Trichoderma longibrachiatum-based biobarriers on nylon sponges where high efficiency of PAH removal was achieved.

29.  P12 line 424-427, the period mark should be replaced by a comma and the initial letter of ‘While’ should be in lower case.

 “…. While” > …, while

30.  P12 line 432, remove “respectively’

31.  P12 Line 462, remove “through’

32.   P13 line 507, including > includes

Author Response

Along with my coauthor, I sincerely thank the esteemed editor/reviewers for their critical/insightful comments made to improve the manuscript. Please find the point-by-point reply to the comments. All the suggested comments were addressed and changes were highlighted in the revised manuscript. We believe that the revised manuscript has been considerably improved now.

Reviewer 1

Major comments

Major issues:

  1. The title may need revising. To the best of our knowledge, “accomplish (achieve) xxx goal”, rather than “address xxx goal”.

Reply: We thank you for the suggestions. The title has been revised as “Advanced fungal biotechnologies in accomplishing sustainable development goals (SDGs): what do we know and what comes next?

  1. Wording redundancy— for example, in the Abstract section (P1 Line 25-27) there is a paragraph as follows, “Delving into how the advances in fungal biotechnology can contribute to/address sustainable development goals (SDGs), state-of-the-art concepts, emerging trends, achievements and prospects/directions in the future, are discussed”, however, at the end of the Introduction section, P3 line 112-114, the author wrote “Delving into how the advances in fungal biotechnology can contribute to sustainable development goals (SDGs), state-of-the-art concepts, transformative approaches, achievements, and prospects/directions in the future, are discussed.”

There are a number of repetitious expressions in every sections throughout this manuscript. For instance, in my opinion, the first paragraph and 2nd paragraph of the Introduction section may be merged to one.

Reply: The authors appreciate the insightful comments. The paragraph “Delving into how the advances in fungal biotechnology can contribute to/address sustainable development goals (SDGs), state-of-the-art concepts, emerging trends, achievements and prospects/directions in the future, are discussed has been modified to reduce word redundancy in the abstract section.

We have thoroughly revised the manuscript to minimize repetitious expression of words in every section. Some of the literature was deleted and modified to maintain clarity. In the introduction section, first paragraph and 2nd paragraph was merged to one. Please refer to the revised manuscript for changes.

  1. Table 1—Representative examples of high-value products from fungi (natural and engineered strains) and their potential to achieve Sustainable Development Goals (SDGs). As for S. cerevisiae, there are three places to exemplify its use in lycopene (carotenoid), ethanol, and fatty acids-derived biofuels, respectively. May these 3 cases be merged into one, under the fungal species “S. cerevisiae”? Same things for A. niger.

Reply: Table 1 has been extensively modified as per the suggestions. Under S. cerevisiae, all the representative studies were included in one section, similarly for A. niger and other species. Please refer to the revised manuscript.

However, as another reviewer suggested to divide the table 1 on the basis of biotechnological applications, sometimes it was not possible to merge different applications under one fungi and were kept under their respective applications.

  1. P 6, Figure 2—Development of a bio-based economy via production and utilization of high-value products from Fungi: The Fig. 2 in this manuscript is totally same as the Figure 1 published in Microorganisms 2023, 11, 1141 by Tiwari and Dufosse. A screenshot of Fig. 1 by Tiwari and Dufosse (2023) is attached below.

To the best of mu knowledge, the admission from the original authors (Tiwari and Dufosse, 2023) /or from the publication journal should be authorized depending on the relevant copyright regulations.

Reply: We are thankful for the suggestion. The figure has been reused from our own manuscript published in MDPI Microorganisms in 2023. We have properly acknowledged the paper and cited the text.

As per the journal policies

MDPI Open Access Information and Policy

All articles published by MDPI are made immediately available worldwide under an open access license. This means:

  • everyone has free and unlimited access to the full-text of all articles published in MDPI journals;
  • everyone is free to re-use the published material if proper accreditation/citation of the original publication is given;

Permissions

No special permission is required to reuse all or part of article published by MDPI, including figures and tables. For articles published under an open access Creative Common CC BY license, any part of the article may be reused without permission provided that the original article is clearly cited. Reuse of an article does not imply endorsement by the authors or MDPI.

Detail comments

Minor issues:

  1. In the Abstract, P1 line 19, when the abbreviation SDGs was initially mentioned, it should be described in full like sustainable development goals (SDGs).

Reply: In abstract, Pi line 19- abbreviation SDG was described in full, sustainable development goals (SDGs

  1. P1 line 32, depleting –> exhausting

Reply: P1 line 32, depleting was replaced with exhausting

  1. P1 line 39,  With significant association and influence on human lives –> with significant association with and influence on human lives

Reply: P1 line 39, With significant association and influence on human lives changed to with significant association with and influence on human lives

  1. P2 line 46-47, expanded for commercial use in food components, the chemical and pharmaceutical sector –> expanded tocommercial use in food additives, the chemical and pharmaceutical sectors.

Reply: P2 line 46-47, expanded for commercial use in food components, the chemical and pharmaceutical sector was revised as expanded to commercial use in food additives, the chemical and pharmaceutical sectors

  1. P2 line 48, to remove ‘being’

Reply: P2 line 48, being word was removed.

  1. Please rewrite the paragraph of P3 line 91-93: “In the last decade, the rapid expansion in fungal biology translated into cascading biomass-conversion technologies, unlocking full potential and facilitating the production of high-value substances as food and feed components.”

Reply: The paragraph of P3 line 91-93 was rewritten for clarity.

  1. P3 line 112-113, to remove the full term for SDGs since it has been defined previously.

Reply: P3 line 112-113, full form of SDG was removed and likewise in the entire manuscript. Please see the changes.

  1. P4 line 122, realizing –> achieving

Reply: P4 line 122, realizing was replaced with achieving

  1. P4 line 128-129, to tackle climate change –> to tackle global climate change

Reply: P4 line 128-129, to tackle climate change was changed, to tackle global climate change

  1. P6 line 136-138, please rewrite the sentence starting with “Fungal biotechnology and bio-based products”. I would suggest the authors decrease the wording/expression redundancy.

Reply: P6 line 136-138, The sentence was revised and rewritten to reduce wording/expression redundancy.

The development of bio-based products via fungal biotechnologies defines potential in tackling hunger and malnutrition, and ensuring food security.

  1. P6 line 138-139, managed –> tackled

Reply: P6 line 138-139 managed was changed to tackled

P6 Line 140, achieved key recognition–> gained key consensus

Reply: P6 Line 140, achieved key recognition was changed to gained key consensus

P6 line 153, time immemorial –> prehistory time;

Reply: P6 line 153, time immemorial was revised as prehistory time;

P6 line 153-154, food products should include cheese, bread, food flavors, etc.

Reply: P6 line 153-154, the sentence was revised as Since prehistoric time, fungal species have been used to prepare beverages and food products including cheese, bread, food flavors etc.

  1. P6 line 157-159, in the sentence of “Discussing bio-based components, edible mushrooms from phyla Ascomycota and Basidiomycota are widely utilized in food preparations across the globe.”. Do the authors express that the “discussing bio-based components, edible mushrooms…are utilized in the global food preparations” or  “bio-based components, edible mushrooms…. are widely utilized in food preparations across the globe.”

I would suggest the word ‘discussing’ should be removed.

Reply: P6 line 157-159, the sentence was revised for clarity. In the sentence, authors express that…. bio-based components, edible mushrooms…. are widely utilized in food preparations across the globe.”

The edible mushrooms from phyla Ascomycota and Basidiomycota are widely utilized in food preparations across the globe and relished as exquisite delicacies.

  1. P7 line 168-172, Please rewrite the sentence starting with “Key studies have documented…”

Reply: The sentence was modified as Research into harnessing the socio-economic benefits of fungi has delved into developing food components comprising nutraceuticals, functional food products, pharmaceuticals, and enzymes. The next sentence starts with Key studies have documented….

  1. P7 line 189, to achieve –> to meet

Reply: P7 line 189, to achieve is changed, to meet

  1. P7 line 190, method –> approach

Reply: P7 line 190, method, has been replaced with approach

  1. P7 line 192, new/novel–> novel

Reply: P7 line 192, new/novel with novel

  1. P7 Line 192-193, achieve –> maintain

Reply: P7 Line 192-193, achieve has been replaced with maintain

  1. P7 line 199-200, “in the environment from symbionts, and decomposers to pathogens” –>”in the environment ranging from symbionts, decomposers to pathogens”

Reply: P7 line 199-200, sentence revised as in the environment, ranging from symbionts, and decomposers to pathogens.

  1. P7 line 202,  …, deal with –> …, to deal with;

Reply: The repetitive sentence was removed.

P7 line 203,  and tackle –>  and to tackle

Reply: The repetitive sentence was removed.

  1. P9 line 316-318,  “The concept of engineering fungal genomes for size reduction relies on the deletion of non-essential genes, and focus on targeted genes/pathways for morphological changes/growth phenotype and the creation of desired fungal chimeras.” –> The concept of engineering fungal genomes for size reduction relies on the deletion of non-essential genes, and focusses on targeted genes/pathways for morphological changes/growth phenotype and the creation of desired fungal chimeras.

Reply: P9 line 316-318, the sentence was revised as suggested.

  1. P10 line 324, there should be a coordinating conjunction like ‘and’ to link the two sentence.

Reply: P10 line 324, and was added to link the two sentences.

22.P10 line 327, remove the repetitive phrase “in filamentous fungi”

Reply: The repetitive phrase was removed and the sentence was revised.

  1. P10 line 336-339, please change the phrase ‘contribute to’ to another one, such as bring about, result in, lead to, pave the road towards…. Throughout the manuscript,  the phrase of ‘contribute to’ had been used for 14 times.

Reply: P10 line 336-339, the phrase will result in was added. We have tried to minimize the repetitive words in the manuscript and replaced with alternate words.

  1. P10 line 351-352,  potent role –> potency

Reply: P10 line 351-352, potent role has been changed to potency

  1. P10 line 354-356,  expression redundancy for “…have great potential and play a key role in…”

Reply: P10 line 354-356, The sentence was modified to reduce expression redundancy.

Mycorrhizal fungi play a key role in the ecosystem, by promoting plant access to nutrients and water in soil and plant tolerance to pathogens.

  1. P10 line 374-382, bioremediation –> bioremediating;

Reply: P10 line 374-382, bioremediation was changed to bioremediating;

textile wastewater can be detoxified by…–> textile wastewater detoxification by…

Reply: The sentence was revised…. textile wastewater detoxification by…

  1. P10 line 384-387, please rewrite the sentence starting with “Fungi classified into…”

Reply: P10 line 384-387, the sentence was revised.

  1. P10 Line 399-401, “advanced approaches for PAH mitigation include Trichoderma longibrachiatum-based biobarriers on nylon sponges, high efficiency of PAH removal was achieved” …–> advanced approaches for PAH mitigation include Trichoderma longibrachiatum-based biobarriers on nylon sponges where high efficiency of PAH removal was achieved.

Reply: P10, Line 399-401, the sentence was modified….. advanced approaches for PAH mitigation include Trichoderma longibrachiatum-based biobarriers on nylon sponges where high efficiency of PAH removal was achieved.

  1. P12 line 424-427, the period mark should be replaced by a comma and the initial letter of ‘While’ should be in lower case.

 “…. While” –> …, while

Reply: P12 line 424-427, necessary changes were made as suggested.

  1. P12 line 432, remove “respectively’

Reply: P12 line 432, “respectively’ word was removed.

31.P12 Line 462, remove “through’

Reply: P12 Line 462, the word “through’ was removed.

  1. P13 line 507, including –> includes

Reply: P13 line 507, including was changed to includes

Reviewer 2 Report

The topic of the review seems interesting, but I have a number of comments.

1. Table 1 should be corrected. The rows «S. cerevisiae” and “A. niger” occur several times in it, the names of some strains are unreasonably abbreviated (“L. edodes”). The last line needs to be detailed. It is necessary to somehow systematize the rows inside the table: either in accordance with the practical application (medical, food, etc.), or in accordance with the systematic position of the fungi. Is it possible that only P. camemberti is used in cheese production? The authors should also add to this table fungi of the genus Beauveria producing insecticides and others that are used as biological agrochemicals. Also, the strains used in the production of fermented foods such as soy sauce, miso and others are not mentioned in any way.

2. The review is very verbose, but not very clearly organized. It seems logical that each section is devoted to one sustainable development goal and what contribution the fungi can make to this. If the authors hold the same view, it is better to indicate this more clearly in the titles of the sections.

Author Response

Reviewer 2

The topic of the review seems interesting, but I have a number of comments.

Detailed comments

  1. Table 1 should be corrected. The rows «S. cerevisiae” and “A. niger” occur several times in it, the names of some strains are unreasonably abbreviated (“L. edodes”). The last line needs to be detailed. It is necessary to somehow systematize the rows inside the table: either in accordance with the practical application (medical, food, etc.), or in accordance with the systematic position of the fungi. Is it possible that only P. camemberti is used in cheese production? The authors should also add to this table fungi of the genus Beauveria producing insecticides and others that are used as biological agrochemicals. Also, the strains used in the production of fermented foods such as soy sauce, miso and others are not mentioned in any way.

Reply: We appreciate the key suggestions for the correction of Table 1.

The rows under fungi Saccharromyces and Aspergillus were combined together, similarly for other fungi. L. edodes was abbreviated since full form was used before in the table. The last line was detailed and highlighted.

The table has been throughly revised on the basis of biotechnological applications (medicine, food, agriculture etc.).

Together with P. camemberti, other cheese producing fungal strains were added. The authors would like to state that there are many examples of fungal resources in diverse biotechnological applications, however, we have discussed only the key literature examples to highlight the prospects of fungi in achieving SDGs.

The table was revised to include fungi of the genus Beauveria producing insecticides and others that are used as biological agrochemicals.

In the food section (food applications of fungi), fungal strain Aspergillus oryzae in soy sauce production, and Miso production was included. Please refer to the revised manuscript for changes.

  1. The review is very verbose, but not very clearly organized. It seems logical that each section is devoted to one sustainable development goal and what contribution the fungi can make to this. If the authors hold the same view, it is better to indicate this more clearly in the titles of the sections.

Reply: The authors are grateful for the suggestions. The manuscript has been extensively revised for improved clarity and organization. Some sentences were shortened.

Many repetitive sentences/words were deleted, and modified as per the suggestions, the sub-titles were revised to indicate clearly the role of fungi in achieving multiple SDGs. We believe that these changes are much needed to clearly define and discuss the theme of the manuscript.

Also, the manuscript has been thoroughly revised for English language, grammar and punctuations. We hope now it has been considerably improved for consideration.

Round 2

Reviewer 1 Report

It is an interesting and comprehensive review paper regarding fungal biotechnology and achieving the UN sustainable development goals (SDGs). Thank the authors for their efforts to revise the manuscript by addressing the reviewer's comments point-by-point. It is acceptable for publishing in JoF, in my opinion. 

Just a point regarding the formatting of subtitle- 2.3 Novel fungal cell factories for the production of bioactive metabolites, which should be in non-italic. 

Reviewer 2 Report

the reviced manuscript can be acsepted

the reviced manuscript can be acsepted